# Benefits and Harms of Treatment and Preventive Interventions for Hereditary Angioedema: Protocol for a Systematic Review and Network Meta-Analysis of Randomized Controlled Trials

**DOI:** 10.3390/genes13050924

**Published:** 2022-05-22

**Authors:** Mati Chuamanochan, Sutthinee Phuprasertsak, Puncharas Weesasubpong, Chidchanok Ruengorn, Chabaphai Phosuya, Ratanaporn Awiphan, Brian Hutton, Kednapa Thavorn, Jonathan A. Bernstein, Surapon Nochaiwong

**Affiliations:** 1Division of Dermatology, Department of Internal Medicine, Faculty of Medicine, Chiang Mai University, Chiang Mai 50200, Thailand; 2Pharmacoepidemiology and Statistics Research Center (PESRC), Chiang Mai University, Chiang Mai 50200, Thailand; puncharas.wee@gmail.com (P.W.); chidchanok.r@cmu.ac.th (C.R.); chaba.pharmacy@gmail.com (C.P.); ratanaporn.a@cmu.ac.th (R.A.); kthavorn@ohri.ca (K.T.); 3Maharaj Nakorn Chiang Mai Hospital, Chiang Mai 50200, Thailand; sutthinee526013@gmail.com; 4Department of Pharmaceutical Care, Faculty of Pharmacy, Chiang Mai University, Chiang Mai 50200, Thailand; 5Ottawa Hospital Research Institute, Ottawa Hospital, Ottawa, ON K1H 8L6, Canada; bhutton@ohri.ca; 6ICES uOttawa, Ottawa, ON K1Y 4E9, Canada; 7School of Epidemiology and Public Health, Faculty of Medicine, University of Ottawa, Ottawa, ON K1G 5Z3, Canada; 8Allergy Section, Division of Immunology, Department of Internal Medicine, College of Medicine, University of Cincinnati, Cincinnati, OH 45267, USA; BERNSTJA@ucmail.uc.edu

**Keywords:** complement 1 inhibitor deficiency, effectiveness, genetic disease, harms, hereditary angioedema, systematic review, treatments

## Abstract

Background: Hereditary angioedema (HAE) is a rare genetic disease that can lead to potentially life-threatening airway attacks. Although novel therapies for HAE treatment have become available over the past decades, a comparison of all available treatments has not yet been conducted. As such, we will perform a systematic review and network meta-analysis to identify the best evidence-based treatments for the management of acute attacks and prophylaxis of HAE. Methods: This study will include both parallel and crossover randomized controlled trials that have investigated prevention or treatment strategies for HAE attacks. We will search electronic databases, including Medline, Embase, PubMed, Cochrane Library, Scopus, and CINAHL, from inception with no language restrictions. Potential trials will be supplemented through a gray literature search. The process of study screening, selection, data extraction, risk-of-bias assessment, certainty assessment and classification of treatments will be performed independently by a pair of reviewers. Any discrepancy will be addressed through team discussion. A two-step approach of pairwise and network meta-analysis will be performed. The summarized effect estimates of direct and indirect treatment comparisons will be pooled using DerSimonion–Laird random-effects models. The incoherence assumption, in terms of the consistency of direct and indirect effects, will be assessed. An evidence-based synthesis will be performed, based on the magnitudes of effect size, evidence certainty, and ranking of treatment effects, with respect to treatment benefits and harms. Discussion: This systematic review and network meta-analysis will summarize evidence-based conclusions with respect to the ratio of benefits and harms arising from interventions for the treatment of acute attacks and prophylaxis of HAE. Evidence from this network estimate could promote the rational use of interventions among people living with HAE in clinical practice settings. PROSPERO registration number: CRD42021251367.

## 1. Introduction

Hereditary angioedema (HAE) is a rare genetic disease that can lead to potentially life-threatening airway attacks [1]. Based on the nature of the disease, HAE results in random, simultaneous, and often erratic symptoms, characterized by attacks of cutaneous and submucosal swelling. HAE is caused by mutations in the C1 inhibitor (C1-INH) gene, serpin family G member 1 (SERPING1), which regulates multiple proteases involved in the complement and contact system, and coagulation and fibrinolytic pathways [2]. The fundamental abnormality in HAE types I and II is due to either a deficit of C1-INH or dysfunctional C1-INH, respectively [2,3]. Another type of HAE, with normal C1-INH function (HAE nC1-INH), was formerly referred to as type III [3].

The combined prevalence rate of HAE types I and II has been estimated to be 1.07 to 1.56 per 100,000, with HAE type I being the most prevalent form [4]. However, HAE nC1-INH is much less prevalent than HAE types I and II, and its true prevalence remains unknown [3]. HAE has a profound effect on patients’ health-related quality of life (HRQOL) because the swelling that accompanies it can cause severe pain and impede patients from using their hands or from walking. Attacks may also limit patients’ ability to perform activities of daily living, including attending school or work [5,6,7,8,9].

Although effective treatment options have become available for HAE over the past decades, a comparison of interventions in a single framework for patients with HAE has not been conducted. To rank the best evidence-based interventions and inform decision-making for patients, clinicians, and policymakers, we aim to conduct a systematic review and network meta-analysis of published randomized controlled trials (RCTs) to identify the most effective options for HAE management. Moreover, a contextualized clinical and methodological approach will be employed to draw evidence-based conclusions regarding the ratio of the benefits and harms of treatment interventions for people living with HAE.

## 2. Materials and Methods

Our systematic review and network meta-analysis will be performed in accordance with the Cochrane Collaboration Handbook for Systematic Reviews of Interventions Version 6.2 [10]. The pre-specified protocol has been registered in the International Prospective Register of Systematic Reviews (PROSPERO) and is available online (http://crd.york.ac.uk/prospero: accessed on 4 May 2022), registration number, CRD42020196592). The present systematic review and network meta-analysis protocol followed the guidelines of the Preferred Reporting Items for Systematic Review and Meta-Analysis Protocols (PRISMA-P) statement [11].

### 2.1. Data Sources and Search Strategy

We will develop an electronic database search strategy, in collaboration with an experienced medical librarian, that will include searching Medline, Embase, PubMed, Cochrane Library (CENTRAL), Scopus, and CINAHL databases from inception, with no language restrictions. The search strategy will be constructed based on a combination of Medical Subject Headings terms or main keywords regarding HAE (i.e., “hereditary angioedema” OR “complement c1 inhibitor protein” OR “edema, hereditary angioneurotic”). Apart from medical conditions, search terms related to types of treatment interventions, based on individual interventions or pharmacological drug classes, will also be used in the search. The pilot pre-specified literature search for each electronic database is provided in Appendix A. We will also search for gray literature from the clinical trial register, Google Scholar, and preprint reports (Appendix A). In addition, potential trials will be supplemented with other eligible trials by searching the reference lists of the retrieved trials, prior systematic reviews, relevant guidelines, and conference meetings from major international dermatology, allergy, and immunology congresses.

### 2.2. Study Selection Process, Eligibility Criteria, and Predefined Outcomes of Interest

A pair of reviewers (S.P. and P.W.) will screen titles and abstracts identified by the literature search and the included records will be subsequently screened for potentially relevant full-text articles to establish the final set of included studies. Any discrepancy will be resolved through discussion/consultation with a clinician (M.C.) and a methodologist (S.N.). Potentially eligible studies in non-English languages will be translated before full-text appraisal. We will include both parallel and crossover RCTs that investigate the safety and efficacy of prophylaxis and treatment of acute attacks among pediatric, adolescent, and adult participants diagnosed with HAE. Key elements of the study design, eligibility criteria, and predefined outcomes, based on the population, intervention, comparison, outcome, timing, and setting (PICOTS) framework, are described in Table 1. For the companion trials, or post hoc analysis studies, we will assemble the relevant information regarding overlapping participants and/or study periods. The pre-specified possible network intervention nodes included in this systematic review and network meta-analysis are presented in Table 2.

### 2.3. Data Extraction

Two reviewers (S.P. and P.W.) will independently extract the pre-specified information using a standardized approach and predesigned electronic extraction form implemented in Microsoft Excel to gather data as follows:Trial characteristics (first and corresponding author’s name, study population [e.g., acute attacks or prophylactic treatment], study design, study setting, country enrollment, sample size, study treatment follow-up period.Participant characteristics (i.e., age of study participants [mean or median, or pre-specified age groups; <18, 18 to <65, and ≥65 years], age at symptom onset, proportion of female participants, race/ethnicity [White, Black, Hispanic, Asian, other), body weight/body mass index, HAE type [type I, type II, or HAE nC1-INH], number of attacks of angioedema before the screening, locations affected by attacks [upper airway, gastrointestinal tract of abdomen, genitourinary, facial, extremity/peripheral], severity of attacks experienced by the patient [i.e., moderate, severe, or severity symptoms score], history of laryngeal attacks, history of allergy, history of first-degree relative with HAE, history of psychiatric disorders or other systemic diseases, baseline complement-related variables [functional C1-INH, antigenic C1-ING, C1q, C4], laboratory results, and concomitant medications).Specific treatment intervention and comparison group, including individual treatment comparisons, a specific dosage of treatment, indications for use (prophylaxis or treatment), route of administration, concomitant and rescue treatment medications.Predefined outcomes, including specific details of the measurement tools used to assess the outcomes of interest.

If quantitative data are not available, we will then digitize the published figures using WebPlotDigitizer 4.4 (https://automeris.io/WebPlotDigitizer, accessed on 4 May 2022) to extract specific numerical values. Where the mean and standard deviation (SD) of the continuous outcome of interest is not provided, we will calculate the sample mean and its SD based on the sample size, median, range, and/or interquartile range as described elsewhere [12]. For any continuous outcomes, we will estimate the treatment effect using the mean change from baseline to the endpoint to overcome the different metric measurements across included trials. If data were not reported, we will calculate the treatment effect for mean change using the formula: ∆value_change_ = value_endpoint_ − value_baseline_, in which SD^2^_change_ = [SD^2^_baseline_ + SD^2^_endpoint_ − (2 × ρ × SD_baseline_ × SD_endpoint_)], where ρ indicates for the correlation coefficient. If ρ is not available, we will use a value of 0.7 for the correlation coefficient between the baseline and end of treatment follow-up, with equal variances among the intervention and comparison groups. For any binary outcome parameters, a correction by 0.5 cells will be applied for trials that reported zero events [13]. For crossover trials, we will only include information from the period before the start of the crossover.

For trials with incomplete or unclear data, we will contact the corresponding author for further clarification. If the authors do not reply after three contact attempts, we will use the most relevant published data or exclude the trial from our analyses. The final set of data will be cross-checked, verified, and any disagreements addressed by two reviewers (M.C. and S.N.).

### 2.4. Risk of Bias Assessment

Two reviewers (M.C. and S.N.) will independently critically appraise the methodological quality of each included trial based on the Cochrane revised tool Version 2 for risk-of-bias assessment [14]. This tool consists of six bias domains, including the randomization process, deviations from intended interventions, missing outcome data, measurement of the outcome, and selection of the reported result. The overall risk of bias of included trials will then be classified into low-, some concerns-, or high-risk of bias. Any disagreements during this process will be addressed through team discussion.

### 2.5. Data Synthesis

Only peer-reviewed full-text will be considered in our primary analysis. A two-step approach of traditional pairwise and network meta-analysis will be performed. For the first step in pairwise meta-analysis, to account for apparent heterogeneity between studies, the summarized effect estimates of direct treatment comparisons will be pooled using the DerSimonion–Laird random-effects model [15]. Heterogeneity across the included trials will be explored using the Cochran Q test, with a *p*-value of less than 0.10. The degree of inconsistency will be assessed using the *I*^2^ index and *τ*^2^ statistics. The degree of inconsistency will be categorized as low (*I*^2^ = 20.0%; *τ*^2^ = 0.04), moderate (*I*^2^ = 50.0%; *τ*^2^ = 0.16), and high (*I*^2^ = 75.0%; *τ*^2^ = 0.36). Publication bias will be assessed using Begg’s and Egger’s tests and visualized funnel plots, with a *p*-value of less than 0.10 in analyses including 10 or more included trials [16,17].

The second step, a frequentist network meta-analysis of aggregate data, will be performed to establish network effect estimates for each outcome of interest using the restricted maximum likelihood method. To provide the highest generalizability and more conservative estimated treatment effects, a random-effects model will be employed to incorporate direct and indirect treatment comparisons. The surface under the cumulative ranking curve (SUCRA) will be calculated for each treatment comparison to rank the hierarchy of treatment interventions in the network estimates [18]. Higher SUCRA scores will indicate a higher ranking for effectiveness (better treatment outcomes, i.e., greater change in symptom scores for treating acute attack or reduction in angioedema attacks for prophylactic treatment) and undesirable treatment (higher risk of all-cause study dropouts [unacceptability of treatment], serious adverse events, or all adverse events).

Sufficiently similar treatment interventions, in terms of transitivity assumptions, will be explored, and the distribution of participants and study characteristics across all included trials (i.e., mean participant age, proportion of female participants, type of HAE, race/ethnicity, and mean body weight/body mass index) will be examined. The incoherence assumption, in terms of consistency of direct and indirect effects, will be assessed using (i) a loop-specific approach, in which inconsistency in every closed loop of evidence will be assessed, and (ii) a design-by-treatment interaction model, in which inconsistency will be assessed through all possible sources in the network, conjointly [19]. Comparison-adjusted funnel plot symmetry will be visualized to assess potential small study effects [18]. Furthermore, 95% prediction intervals (for all network meta-analysis estimates) will also be estimated, which will account for the predicted range for the true treatment effect in an individual study [20].

Preplanned subgroup analyses will be performed based on the following: (i) age of study participants (pediatric/childhood or adolescent [<12 years] versus non-elderly adults [18-65 years] versus older adults [≥65 years]); (ii) sex (as reflected by the proportion of female participants); (iii) study size (<50 vs. ≥50 participants); (iv) study design (parallel versus crossover RCTs); (v) HAE type (i.e., type I, type II, or HAE nC1-INH); and (vi) geographical region of study recruitment (i.e., North America, Europe, or other regions). Network meta-regression analyses will be performed by supplementing other covariates, as mentioned in the adjusted network meta-analysis model. Moreover, a set of sensitivity analyses will also be applied for all outcomes in the further analysis to address the robustness of our findings including (i) adding trials that only provide an abstract or unpublished or preprint results; (ii) restricting the analysis to only trials with parallel group design; (iii) excluding trials with high overall risk-of-bias assessment; and (iv) assuming a ρ value of 0.5 in cases with missing correlation coefficients for continuous outcomes.

The effect estimates of traditional pairwise meta-analysis and network meta-analysis will be expressed as standardized mean differences (SMDs) and odds ratios (ORs) along with the 95% confidence intervals (95% CIs). If there is limited relevant information of the included trials, we will conduct a systematic review and narrative synthesis with respect to the key participant characteristics and treatment comparisons.

Statistical significance for all tests will be two-tailed, with *p*-value < 0.05. All analyses will be performed using STATA software (version 16.0; StataCorp, College Station, TX, USA).

### 2.6. Certainty Assessment and Classification of Treatment Interventions

Two reviewers (MC and SN) will independently grade the certainty of evidence and rate the evidence for each outcome by applying the modified confidence in network meta-analysis (CINeMA) approach [21] and Grading of Recommended Assessment, Development, and Evaluation (GRADE) approach [22], respectively. Overall, the strength of evidence will be categorized into very low-, low-, moderate-, and high-quality evidence (Table 3).

Next, based on clinical and methodological points of view, we will employ a contextualized approach to establish the treatment network effect estimates with respect to the dimension of efficacy (treatment responses/efficacy for the prevention of attacks) and safety profiles (unacceptability of treatment, serious adverse events, and all adverse events). To draw evidence-based conclusions and classify treatment interventions, key components including evidence certainty, SURCA values, and effect size magnitudes will be incorporated into a single framework assessment. Specifically, the magnitudes of effect size will be estimated to be very small (SMDs < 0.2 or ORs < 1.68), small (SMDs = 0.2–0.4 or ORs = 1.68–3.46), medium (SMDs = 0.5–0.7 or ORs = 3.47–6.71), and large (SMD ≥ 0.8 or ORs > 6.71) [23,24]. Taken together, interventions will be classified as having trivial (not different from placebo/standard treatment/usual care), small, moderate, and large effects to inform clinical interpretation and rank the clinical evidence of the findings.

## 3. Ethics and Dissemination

Based on the nature of the systematic review and network meta-analysis study using existing published information, ethical approval is not required. This study involved no individual patient information; thus, informed consent is not required. Our findings will be reported in line with the Preferred Reporting Items for Systematic reviews and Meta-Analyses (PRISMA) 2020 statement guidelines [25] along with the PRISMA extension statement for reporting of systematic reviews incorporating network meta-analyses of healthcare interventions [26]. Our findings will be presented through scientific meetings and published in peer-reviewed journals. Any amendments to this protocol will be described in the final report.

## 4. Discussion

Before 2009, there were no US Food and Drug Administration-approved treatment options for acute attacks of angioedema. Treatments available were generally limited to supportive care and efforts to protect the airway [9]. Over the past decades, several new treatment interventions have become available and been approved for acute attacks, and as targeted prophylactic therapy for HAE with C1 inhibitor deficiency [2]; however, there continues to be an unmet need for safe and effective treatments for HAE [5,6,7,8]. Apart from being a rare and life-threatening disease, living with HAE also places a significant burden on patients’ families, caregivers, healthcare providers and system, and society. Therefore, it is important to have a better understanding of the most effective HAE treatments and prophylactic interventions.

Given diverse study populations, our findings may be underrepresented in some racial or ethnicity parameters, which could limit the generalizability of the study findings. Generally, evidence from clinical trials in several settings almost solely represents the White population. To the best of our knowledge, minority patients living with HAE are underrepresented in clinical trials, and, in particular, those who are at risk for additional disease burden. Evidence suggests that Hispanic patients are underdiagnosed with HAE. Furthermore, disparities in treatment practice and therapeutic interventions have been observed among White and Black HAE patients [27].

This study will summarize all available evidence on both acute attacks and prophylactic treatments for HAE. Our study will be performed based on a comprehensive and rigorous approach with no language restrictions. We will ensure that our findings have a positive effect on the health outcomes of people living with HAE, including their families and caregivers. Moreover, findings from the evidence-based synthesis will also facilitate identifying the best treatment options for HAE care management for healthcare providers, policymakers, researchers, and public society. The study findings will have the potential to influence and inform international guidelines on acute attacks and prophylactic treatment for people living with HAE.

## 5. Conclusions

This systematic review and network meta-analysis will systematically summarize and compare all available evidence on the safety and efficacy of treatment and prevention options for HAE. Contemporary evidence from this network estimate could promote the rational use of interventions in clinical practice. Our findings will be disseminated in international scientific conferences and a formal peer-reviewed publication.

## Figures and Tables

**Table 1 genes-13-00924-t001:** The PICOTS: study inclusion/exclusion criteria.

Category	Criteria for Inclusion	Criteria for Exclusion
Populations	Pediatric, adolescent, or adult patients diagnosed with HAE, including HAE with deficit C1-inhibitor levels, HAE with dysfunctional C1-inhibitor, and HAE with normal C1-inhibitors function	Studies recruiting participants with an unclear definition of HAE or other form of angioedema Studies including less than 10 participants with HAE In vitro or animal studies
Interventions	Treatment interventions options for HAE with any type of administered dosage treatment for acute attacks or prophylactic treatment	Studies with the disconnected node of treatments
Comparators	Placebo, active comparator, or standard of care	Studies without control groups
Outcomes	Primary outcomes for acute attacks treatment ○ Time-to-relief: time from the start of treatment to onset of symptom relief ○ Time-to-resolution: time to complete resolution of HAE symptoms ○ Treatment response: percentage of patients reporting significant improvement ○ Change in symptoms score and treatment outcome score ○ Unacceptability of treatment (all-cause study dropouts) Primary outcomes for prophylactic treatment ○ Number of angioedema attacks ○ Number of attacks requiring acute treatment ○ Number of moderate or severe attacks ○ Percentage of patients who had a response to treatment: reduction of 50% or more in the number of attacks ○ Unacceptability of treatment (all-cause study dropouts) Secondary outcomes (both acute attacks and prophylactic treatment) ○ Percentage of attack-free patients ○ Number of attack-free days ○ Number of high-morbidity attacks ○ Health-related quality of life and other PROs ○ Serious adverse events ○ Discontinuation of treatment due to adverse events/serious hospitalization sequelae ○ Incidence of all adverse events ○ Treatment failure: lack of efficacy or need for rescue treatment ○ Healthcare utilization and costs	Studies not providing data to calculate the efficacy or safety of outcomes of interest
Timing	An extensive search strategy from the inception of bibliographic databases forward to assure all published literature was identified	No limit timing of start date No language restriction
Setting	Experimental study: RCTs (parallel or crossover trials)	Non-randomized studies, observational studies (cohort, case-control, and cross-sectional studies), N-of-one trials, case series/case reports, reviews, and systematic review and meta-analysis

Abbreviations: HAE, hereditary angioedema; PICOTS, population, intervention, comparison, outcome, timing, and setting; PROs, patient-reported outcomes; RCTs, randomized controlled trials.

**Table 2 genes-13-00924-t002:** Pre-specified possible network intervention nodes for HAE treatments.

Treatment of Acute Attacks of HAE	Prophylactic Treatment of HAE
C1 esterase inhibitor (C1-INH): i.e., plasma-derived nano-filtered C1-INH (Cinryze^®^), plasma-derived C1-INH (Berinert-P^®^ or Haegarda^®^), recombinant human C1 inhibitor (Ruconest^®^) Bradykinin-B2-receptor antagonist: i.e., icatibant, PHA022121^®^ Fresh frozen plasma Solvent detergent plasma Kallikrein inhibitors: i.e., ecallantide, berotralstat (BCX7353), KVD900^®^	Attenuated androgens: i.e., danazol, stanozolol, oxandrolone, methytestosterone Anti-fibrinolytic agents: i.e., epsilon aminocaproic acid (EACA), tranexamic acid C1-INH concentrate: i.e., plasma-derived nano-filtered C1-INH (Cinryze^®^), plasma-derived C1-INH (Berinert-P^®^) Kallikrein inhibitors: i.e., berotralstat (BCX7353), lanadelumab, ATN-249^®^, KVD824^®^ RNA interference targeted at FXII: i.e., ALN-F12^®^, ARC-F12^®^ Adeno-associated virus antibody delivery gene therapy: i.e., BMN 331^®^ Humanized anti-FXIIa monoclonal antibody: i.e., garadacimab^®^ Antisense targeting prekallikrein: i.e., IONIS-PKK-LRx^®^ Bradykinin-B2-receptor antagonist: i.e., PHA022121^®^ CRISPR/Cas9 editing of KLKB1: i.e., NTLA-2002^®^

Abbreviations: HAE, hereditary angioedema.

**Table 3 genes-13-00924-t003:** Modified criteria for certainty assessment based on CINeMA and GRADE approach [21,22].

Judgement	Criteria	Instruction for Downgrading
Within-study bias	Within-study bias will be evaluated by majority of risk of bias assessment results within each comparison We will increase the concern to one level for comparisons with single study only	Major concerns: downgrade the evidence one level Some concerns: downgrade the evidence one level with 2 or more some concerns in other judgements
Reporting bias	Reporting bias will be evaluated by non-statistical consideration of likelihood of non-publication of evidence We will increase the concerns to one level for outcomes with evidence of small study effects in the network by comparison adjusted funnel plot	Major concerns: downgrade the evidence one level Some concerns: downgrade the evidence one level with 2 or more some concerns in other judgements
Indirectness	Populations among studies will be assessed by distributions of age, gender, and comorbidities For continuous outcomes (symptoms score), outcomes assessment within each comparison will be evaluated by the directness of measurement tool	Major concerns: downgrade the evidence one level Some concerns: downgrade the evidence one level with 2 or more some concerns in other judgements
Imprecision	Imprecision will be focused on width of CI based on a clinically important mean difference of 0.2 for continuous outcomes (urticarial symptoms, pruritus severity, and hives severity) and odds ratio of 1.2 for binary outcomes (unacceptability of treatment, serious adverse events, and all adverse events) We will increase the concern to one level if the width of CI is between 4 times and 10 times of lower limit The concern level will increase two levels if the width of CI is above 10 times of lower limit	Major concerns: downgrade the evidence one level Some concerns: downgrade the evidence one level with 2 or more some concerns in other judgements
Heterogeneity	Heterogeneity will be evaluated according to the CINeMA documentation by variability of effects in relation to the clinically important size of effect and between-study variance for the network meta-analysis We will increase the concern to one level if there is no information regarding between-study heterogeneity for each direct comparison or I2 index >60% in the direct comparison or inconsistency between 95% CI and 95% PrI	Major concerns: downgrade the evidence one level Some concerns: downgrade the evidence one level with 2 or more some concerns in other judgements
Incoherence	Incoherence will be evaluated by the design-by-treatment intervention model globally and the loop specific approach We will increase the concern to one level if there is evidence of incoherence in the agreement between the main analysis and a set of sensitivity analyses	Major concerns: downgrade the evidence one level Some concerns: downgrade the evidence one level with 2 or more some concerns in other judgements
**Quality of the evidence (GRADE):** ○ **High quality:** further research very unlikely to change our confidence in the estimate of effect ○ **Moderate quality:** further research likely to have an important impact on our confidence in the estimate of effect and may change the estimate ○ **Low quality:** further research very likely to have an important impact on our confidence in the estimate of effect and is likely to change the estimate ○ **Very low quality:** very uncertain about the estimate		

Abbreviations: CI, confidence interval; CINeMA, Confidence In Network Meta-Analysis; GRADE, Grading Recommendations Assessment, Development and Evaluation; PrI, prediction interval.

## Data Availability

Data sharing is not applicable to this article as no datasets were generated or analyzed during the current study. However, data from the study will be made available at the end of the study, on request.

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
