# Peer review of "Benefits and Harms of Treatment and Preventive Interventions for Hereditary Angioedema: Protocol for a Systematic Review and Network Meta-Analysis of Randomized Controlled Trials"

_genes, 2022, doi:10.3390/genes13050924_

Round 1

Reviewer 1 Report

I think this article is well written and complete.

Author Response

Thank you very much for your pertinent observation.

Reviewer 2 Report

Authors planned to address an important issue trying to assess whether can be extrapolated clinical meaningful information from trials for the therapy of patients with HAE.

Overall the manuscript is well-written and the aims clearly identified.

I wonder how the Authors would manage information in the case of a scarcity and heterogeneity of study conducted in this field.

Being HAE a rare disease, it can be conceivable that few clearly firm data would be extrapolated. In addition, it can be anticipated that the vast majority of study were conducted in patients suffering fron C1-INH-HAE belonging to Caucasian settings. How the Authors intend to address these flaws?

Author Response

#1. I wonder how the Authors would manage information in the case of a scarcity and heterogeneity of study conducted in this field.

Thank you very much for your insightful comments. To address your concerns regarding scarcity and heterogeneity information, we will use the random-effects models in all outcomes of interest and stated in the “Data Synthesis” section (Page 5, lines 174-177 and Page 6, lines 184-188) as read:

For the first step in pairwise meta-analysis, to account for apparent heterogeneity between studies, the summarized effect estimates of direct treatment comparisons will be pooled using the DerSimonion–Laird random-effects model [15].

AND

Next, a frequentist network meta-analysis of aggregate data will be performed to establish network effect estimates for each outcome of interest using the restricted maximum likelihood method. To provide the highest generalizability and more conservative estimated treatment effects, a random-effects model will be employed to incorporate direct and indirect treatment comparisons.

#2. Being HAE a rare disease, it can be conceivable that few clearly firm data would be extrapolated. In addition, it can be anticipated that the vast majority of study were conducted in patients suffering from C1-INH-HAE belonging to Caucasian settings. How the Authors intend to address these flaws?

Thank you very much for your pertinent observation. We totally agree with your opinions. Where possible, both pairwise meta-analysis and network meta-analysis will be analyzed to provide the best available evidence. To better understand and make it more clear, we have stated these issues under the “Data Synthesis” (Page 6, lines 222-224) and “Discussion” (Page 8, lines 272-279) sections as follows:

If limited relevant information of the included trials, we will conduct a systematic review and narrative synthesis with respect to the key participant characteristics and treatment comparisons.

AND

Given diverse study populations, our findings may be underrepresented in some racial or ethnicity, which could limit the generalizability of the study findings. Generally, evidence from clinical trials in several settings almost solely represents the White population. To the best of our knowledge, minority patients living with HAE are underrepresented in clinical trials, in particular, who are at risk for additional disease burden. Evidence suggested that Hispanic patients are underdiagnosed with HAE. Furthermore, disparities in treatment practice and therapeutic interventions were observed among White and Black HAE patients [27].

Round 2

Reviewer 2 Report

None.